# OpenReview forum: "AgenticPA: Toward Automated and Large-Scale Prompt Attacks on LLMs"
_ICLR.cc/2026/Conference — ICLR 2026 Conference Withdrawn Submission_

### Official Review · Reviewer_xhDQ · 2025-10-28

**Soundness:** 2
**Presentation:** 3
**Contribution:** 3
**Rating:** 6
**Confidence:** 4

**Summary:**

The paper introduces an agent-based pipeline to standardize and automate the reproduction of jailbreak attack-related papers. The approach involves a Paper Agent to extract attack details from research papers, a Repo Agent to pull implementation specifics from associated GitHub repositories, and a Code Agent that instantiates the attacks to generate harmful, executable prompts against target LLMs. The experiments in the paper involve scanning all CS papers on arXiv, selecting 104 for reproduction, and compiling an unified attack library after reproduction. The authors then analyze performance trends across the reproduced attacks and evaluate the fidelity of their reproductions.

**Strengths:**

- I found the motivation of the paper quite interesting, and the work is also timely. The methods are well motivated and presented.
- The scope of the paper is quite large - it involves processing the entirety of CS ArXiv to identify easily reproducible papers, followed by an agentic approach to unify their reproduction. In the process, this work can also facilitate checking for the effectiveness of older attacks on newer models.
- The transformations introduced in the paper allows for quick validation of jailbreak attacks for systems such as RAG and ReAct by directly recreating the attack prompts that would have been created by these systems and testing them with the target LLM.
- In the process of curating all of CS ArXiv, the paper presents some interesting statistical insights about existing jailbreaking approaches, as well as their reproducibility.
- The AutoPABench artifact is also impressive in its scale, unifying datasets and jailbreaking approaches from prior literature.

**Weaknesses:**

- The biggest gap in this work is the measurement of reproduction quality. It does not check to see if results from the original papers are recreated by the proposed agentic approach, therefore casting doubt about the faithfulness of the recreation.
- The paper reports that extensive manual debugging is often required to correct the codebases generated by the agentic approach, somewhat limiting its practicality. I would also argue that the reproduction time of 22.6 minutes per paper is quite high - for a well-written paper the reproduction simply involves pulling the repo of the paper from Github and running a few commands. Some further analysis around papers that particularly benefited from reproduction through the proposed approach would be good to see.
- The paper is missing some implementation details. For example “Each reproduction is validated using 10 test cases.” in line 288 - how does this validation work? What test cases are used? Section 5 is also heavily underspecified - how are the ASRs in table 3 calculated? What is the underlying dataset used for the evaluation? How are the attacks clustered by category?
- While the paper mentions unification, it does not propose a single red-teaming dataset or evaluation metric to evaluate all of the reproduced attacks, therefore still making it hard to compare them in a single benchmark.
- The appendix, while comprehensive, is not linked well in the main body of the paper, which somewhat hurts clarity.

**Questions:**

- Can you provide a comparison of the breadth of attacks covered by this paper, to that in other existing benchmarks?
- Can this framework support compositions of attacks from different papers? Prior work [1][2] has found that these compositional attacks tend to be much more effective.
- Some further information about the amount of manual labor that went into identifying candidate papers for the pipeline, as well as correcting the generated codebases, would be helpful.
- What percentage of reproduced attacks diverged significantly from the original Github repos?
- The methodology mentions at line 242 that files are relocated based on their extensions - is it possible that these files are used to manage prompts critical for the reproduction?
- What is the “interface-level validation” mentioned in line 367? Is this manual as well?
- How does this approach set hyperparameters for reproduced attacks? The original papers often contain heuristics for setting these hyperparameters, which can be buried in the text - are these captured as well?

[1] h4rm3l: A language for Composable Jailbreak Attack Synthesis. Moussa Koulako Bala Doumbouya, Ananjan Nandi, Gabriel Poesia, Davide Ghilardi, Anna Goldie, Federico Bianchi, Dan Jurafsky, Christopher D. Manning
[2] Jailbroken: How Does LLM Safety Training Fail? Alexander Wei, Nika Haghtalab, Jacob Steinhardt.

---

### Official Review · Reviewer_Dff9 · 2025-10-30

**Soundness:** 3
**Presentation:** 3
**Contribution:** 2
**Rating:** 4
**Confidence:** 4

**Summary:**

The paper introduces AGENTICPA, a three-agent system (Paper, Code, Repo) that automates the reproduction of prompt-based attacks on large language models. It extracts attack logic from research papers and repositories, generates executable scripts, and compiles them into a unified benchmark called AutoPABench. The framework reproduces 104 prior attack studies with an average execution success rate of 93 % and cost of $2 per paper, enabling scalable evaluation of LLM safety. Experiments show that recent models are more robust but still vulnerable to multilingual and indirect-injection prompts.

**Strengths:**

1. The multi-agent framework (Paper, Repo, and Code Agents) is well-designed and demonstrates system-level engineering to automate complex attack reproductions.

2. Enables large-scale, consistent reproduction of 100+ prompt-attack papers, addressing reproducibility and evaluation gaps in LLM safety research.

3. The paper is clearly written and well-organized, making the methodology and results easy to follow.

**Weaknesses:**

1. The definition of prompt-based attacks adopted in this paper seems overly broad, grouping together fundamentally different threat types such as jailbreaks, prompt injections, and prompt extractions. These attack families vary significantly in mechanism, reproducibility, and difficulty. Could the authors provide a more detailed analysis of how AGENTICPA handles each attack category, and clarify whether the framework adapts its reproduction or evaluation strategy accordingly?

2. The paper collection stage still requires final human verification of associated GitHub repositories. It is unclear how human reviewers managed to filter and verify papers from the initial pool of 274,297. How many annotators were involved, what were their agreement rates, and what detailed criteria were used for manual judgment?

3. The proposed framework’s safety layer and rollback mechanism are described conceptually but not quantitatively evaluated. Could the authors report how frequently these mechanisms are triggered and how they affect attack fidelity, success rate, and computational efficiency?

4. A direct comparison with existing frameworks, such as AgentSecurityBench or AutoAdvExBench, is missing. How does AGENTICPA differ from these prior works in terms of scope, scalability, and robustness?

5. The font size in several figures is too small, making them difficult to read. Could the authors improve figure readability or consider alternative visualizations?

**Questions:**

See my weaknesses above.

---

### Official Review · Reviewer_8VbE · 2025-10-31

**Soundness:** 2
**Presentation:** 2
**Contribution:** 3
**Rating:** 4
**Confidence:** 4

**Summary:**

This paper presents AGENTICPA that automates the reproduction of prompt attacks on LLMs from academic research. The method employs a three-agent system: a paper agent extracts attack algorithms and evaluation criteria from publications; a repo agent analyzes associated code repositories for implementation details and handles data files; a code agent then synthesizes this information to write, execute, and debug scripts, creating standardized reproductions of the attacks. The method is then applied to generate the AutoPABench benchmark for assessing LLM safety.

**Strengths:**

* Originality: The paper introduces a new multi-agent method to automate the traditionally manual task of reproducing LLM attacks from academic literature.
* Clarity: The paper clearly articulates the problem, the roles and interactions within its three-agent architecture.
* Significance: The evaluations show that it addresses a major bottleneck in LLM safety research.

**Weaknesses:**

* The code agent synthesizes information from a theoretical paper and a practical code repository. However, the authors do not provide clear principle for how it resolves discrepancies between these two sources. This would create a significant risk of producing the attack that is faithful to neither the original paper's intent nor its actual implementation. This ambiguity undermines the claim of faithful reproduction.

* The method's reliance on papers with accessible code repositories introduces a fundamental selection bias into the resulting AutoPABench. The benchmark does not represent the landscape of all published attacks, but only those that are already easy to reproduce. Conclusions drawn from this benchmark, such as any trends in attack efficacy or the prevalence of certain techniques, are not generalizable to the entire threat landscape. I am afraid this would even present an overly optimistic view of LLM defenses.

* The method's debugging process would be a critical flaw. In its effort to create an executable script, the code agent may alter the original logic of an attack simply to make it run (would this happen?). This prioritizes functionality over scientific fidelity. A successful reproduction might pass a simple input-output check while operating on a different principle than the original attack, which renders it invalid as a true replication and corrupts the integrity of the benchmark.

**Questions:**

* Regarding Weakness 1, what is the agent's hierarchy of truth when a discrepancy arises? Does it prioritize the repository's code, the paper's mathematical formulas, or the paper's descriptive text?

* Regarding Weakness 2, what was the total number of attack papers initially identified for the study? what percentage of those were discarded due to the unusability of a code repository? have the authors performed any analysis on the characteristics of the excluded papers? do they differ systematically from the included papers in terms of attack type, novelty, or publication venue?

* Regarding Weakness 3, what is the precise scope of the code agent's debugging powers? can it alter variable assignments, control flow, core function calls, etc., or is it just limited to some superficial fixes like library imports? how do the authors validate that a debugged script remains a faithful implementation of the original attack? is there any human-in-the-loop verification for reproductions that required code modifications? could the authors provide statistics on what proportion of the 104 successful reproductions required automated debugging, and what were the most common types of modifications the agent made?

---

### Official Review · Reviewer_aPsS · 2025-11-02

**Soundness:** 2
**Presentation:** 3
**Contribution:** 2
**Rating:** 2
**Confidence:** 5

**Summary:**

The authors claim that:
- Most LLM safety evaluation systems are limited in scope and only assess attacks in isolation or at a small scale.
- It is not clear whether frontier models are robustly safe against the "full spectrum" of prompt attacks.
- A proposed formalism mapping (Attack, Attack Mechanism) to (Adversarial Prompt) is sufficient to unify the "full spectrum" of attacks and consistently evaluate such attacks across different LLMs.

The authors propose:
- Agentic Prompt Attack (Agentic PA) implementing the proposed formalism that is comprised of 3 agents:
    - Paper Agent, which extracts attack specifications from research papers
    - Repo Agent, which retrieves the implementation details of attacks from GitHub
    - Code Agent, which generates a set of adversarial inputs for a given attack.

Using their framework, the authors reproduced 104 papers to build a library of adversarial prompts. Evaluating LLMs using these prompts, the authors found that recent frontier models are vulnerable to a wide range of known threats.

**Strengths:**

- Reproduced 104 prompt attack papers to build a large-scale standardized library
- Provided evidence that recent frontier models are vulnerable to a wide range of known threats
- The idea of automatically standardizing jailbreak attacks from research papers and public code repositories is interesting.
- The paper is well formated
- This approach resulted in a library of adversarial prompts that could be useful for future work

**Weaknesses:**

- **The "full spectrum of prompt attacks" is not sufficiently specific**, as the full set of possible prompt attacks is infinite.
- **The claim that large-scale evaluations have not been previously done is not accurate.** E.g. See [2] and [6]
- **The claim that prior work only tests attacks in isolation and at a small scale is not substantiated:**
    - [3] Showed that the composition of individual black box attacks can result in more successful attacks
    - [2] Proposed a formal language to represent and compose attacks, implemented several primitives from the literature, and proposed a framework that synthesizes novel combinations of black-box attacks.
- **The redefinition of the type of attacks covered is not justified.** These are black-box jailbreak attacks and are concretely string transformations, not strings.
    - e.g., the low-resource translation attack translates the input prompt into an under-served language. The attack is the translation function, not its output for a particular input.
- **The unification formalism is unclear and incomplete:**
    - It doesn't distinguish attack mechanisms (e.g. transliterate the prompt in arabic [1]) from illicit requests (e.g., "how to corrupt a mayor?") in such a way that different illicit requests can be processed through an attack mechanism to generate their adversarial version.
    - It doesn't allow computing attack success rates uniformly across multiple attacks as done by [1], [2], [3], [4] and [6] using standardazied databases of illicit requests such as [5]. In these prior works, attack success rates are computed as the average rate of target LLM misbehavior in response to illicit requests transformed by an attack mechanism (a string transformation), enabling the computation of ASRs for various attacks using the same set of illicit requests. The formalism proposed in the present work doesn't seem to enable such evaluation.

- **This work doesn't seem to be sufficiently grounded in prior work**, which has proposed unification formalisms for black-box jailbreak attacks ([3], [2], [6]). These prior works define black-box attacks as string-to-string transformations. Prior work also discussed the composability of black-box attacks and a compositional language for representing and synthesizing black-box attacks. Those prior formalizations and unifications are not discussed in this paper or compared to the proposed approach.
Prior work such as [4] has also evaluated white-box attacks using standardased datasets of illicit requests such as [5]. See Figure 16 in [6] for an illustration of the distinction between attacks and illicit requests (harmful prompts).




# References
- [1] Ghanim, M., Almohaimeed, S., Zheng, M., Solihin, Y., & Lou, Q. (2024, November). Jailbreaking LLMs with Arabic Transliteration and Arabizi. In Proceedings of the 2024 Conference - on Empirical Methods in Natural Language Processing (pp. 18584-18600).
- [2] Doumbouya, M. K. B., Nandi, A., Poesia, G., Ghilardi, D., Goldie, A., Bianchi, F., ... & Manning, C. D. h4rm3l: A Language for Composable Jailbreak Attack Synthesis. In The - Thirteenth International Conference on Learning Representations.
- [3] Wei, A., Haghtalab, N., & Steinhardt, J. (2023). Jailbroken: How does llm safety training fail?. Advances in Neural Information Processing Systems, 36, 80079-80110.
- [4] Li, Qizhang, Yiwen Guo, Wangmeng Zuo, and Hao Chen. "Improved generation of adversarial examples against safety-aligned llms." Advances in Neural Information Processing Systems - 37 (2024): 96367-96386.
- [5] Mazeika, M., Phan, L., Yin, X., Zou, A., Wang, Z., Mu, N., ... & Hendrycks, D. (2024, July). HarmBench: A Standardized Evaluation Framework for Automated Red Teaming and Robust - Refusal. In International Conference on Machine Learning (pp. 35181-35224). PMLR.
- [6] Sharma, M., Tong, M., Mu, J., Wei, J., Kruthoff, J., Goodfriend, S., ... & Perez, E. (2025). Constitutional classifiers: Defending against universal jailbreaks across thousands of hours of red teaming. arXiv preprint arXiv:2501.18837.

**Questions:**

- Why does the Paper Agent encode asset files in base64? Is this purely for the purpose of serializing images into text so that it can be processed through text-to-text LLM interfaces? Is there evidence of reliable LLM processing of base64 encoded images?
- It is unclear how the Repo Agent validates code from repositories.
- How are jailbreak attacks represented? Are they merely a list of adversarial inputs? Or string transformations?
- What are executable attack scripts? Such scripts are not jailbreak attacks, but programs that run a set of "jailbreak prompts."
- Many of the papers listed in Table 9 are not aligned with the proposed definition of "prompt attacks" as aiming to "deliver adversarial prompts to target LLMs." E.g. Black-box jailbreak attacks are transformations (e.g., obfuscation, low-resource translation, or the addition of a prefix-suffix, etc., or combinations thereof) that cause an LLM to generate a harmful response to a prompt for which it would normally issue a refusal response due to safety purposes.  How do the authors unify their proposed formalism and the prior formalism?
- Are the reported ASRs computed using the same set of illicit requests? If not, how is their comparison across attacks justified?
- The authors referenced [2], which released a library of 2,656 jailbreak attacks expressed in a formal language (h4rm3l). How were those integrated in this work?

---

### Note · Authors · 2025-11-29

I have read and agree with the venue's withdrawal policy on behalf of myself and my co-authors.